# Toward Fair and Transparent Vision Transformers: Reproducing FairViT and Introducing FairDeiTA

## Abstract

Vision Transformers (ViTs) have achieved state-of-the-art performance in image recognition but frequently inherit social biases from large-scale training data, raising concerns about fairness and transparency. FairViT was recently proposed to mitigate biases in ViTs through adaptive masking while preserving high accuracy with a distance-based loss. This study reproduces and evaluates FairViT's claims on the CelebA dataset, focusing on accuracy and fairness metrics. Contrary to the original paper's findings, our experiments reveal that FairViT does not outperform the baseline model in both performance and fairness. To enhance transparency, we apply interpretability techniques, including Gradient Attention Rollout (GAR) and local surrogate explanations (Ribeiro et al., 2016), providing deeper insight into the learned representations of FairViT. Our reproducibility study underscores the challenges of implementing and verifying fairness interventions in ViTs. Finally, we propose an adversarial debiasing (Zhang et al., 2018) component that improves fairness metrics while maintaining competitive accuracy, offering an alternative direction for fairness-focused ViT-based applications. We formulate this model as FairDeiTA.

## 1 Introduction

In recent years, Computer Vision has experienced a significant shift in preferred model architectures (Park & Kim, 2022; Xia et al., 2024). While Convolutional Neural Networks once dominated the field, ViTs have emerged as a transformative alternative (Yin et al., 2022). Originally developed for sequence modeling in Natural Language Processing (Vaswani et al., 2017), Transformers offer advantages such as effective parallelization and scalability, especially when trained on large datasets (Dosovitskiy et al., 2020). The growing availability of extensive data and powerful hardware has allowed models with billions of parameters (Yu et al., 2022), leading ViTs to achieve state-of-the-art performance on core image recognition benchmarks (Liu et al., 2023).

Despite their success, ViTs still inherit biases from training datasets, as these datasets reflect human-generated data and associated biases, leading to skewed distributions (Luo et al., 2024; Dixon et al., 2018). Furthermore, Transformers often lack interpretability due to their reliance on global attention (Qiang et al., 2023). To address these challenges, Tian et al. (2024) proposed FairViT, a framework that enhances fairness in ViTs through adaptive masking and distance-based loss. In addition, FairViT improves interpretability via GAR by visualizing the model's attention through a heatmap overlaid on top of a single image. Fairness in the FairViT model is assessed using fairness definitions and evaluation metrics derived from recent research on machine learning fairness (Feldman et al., 2015).

This paper aims to reproduce the original framework of the FairViT model. Additionally, we extend the FairViT model by adding a debiasing component that reduces inherent biases in the model, while also improving transparency in its evaluation. The main contributions are as follows.

1. Reproduced the original FairViT paper and compared the results with its reported findings to assess its reproducibility.

2. Improved the implementation of the original code by fixing unintended behaviors and implementing a training regime for the vanilla model.

3. Implemented their interpretability method and conducted a transparency assessment to provide deeper insight into the model's decision-making.

4. Enhanced fairness in the model's decision-making process by incorporating an additional loss component into the total loss, formulated as adversarial debiasing.

The remainder of this paper is organized as follows. Section 2 presents the scope of reproducibility and the main claims made by the authors of the original paper. Section 3 describes the the FairViT model's architecture. Section 4 outlines our methodology, dataset, proposed extensions, and experimental details. Section 5 discusses the results of reproducing the original paper, and introduces our proposed model FairDeiTA. Finally, Section 6 concludes with key findings and future directions. Throughout the paper, we refer to our GitHub repository for additional details on the implemented extensions, including the refactored version from the FairViT paper. The repository is available at `https://anonymous.4open.science/r/FairDeiTA-7248`.

## 2 Scope of reproducibility

The paper from Tian et al. (2024) extends the original ViT model (Dosovitskiy et al., 2020). It addresses the challenge of achieving fairness in ViTs without sacrificing accuracy, by incorporating two key innovations: adaptive masking and distance loss. Both mechanisms serve distinct purposes: adaptive masking enhances fairness by dynamically modifying attention layers to reduce bias across sensitive groups, while the distance loss acts as a regularizer to improve classification performance.

The main claims of the original paper are:

- *Claim 1*: An adaptive masking framework enhances FairViT's fairness, outperforming alternatives.

- *Claim 2*: A distance loss function adjusts output scores to improve FairViT's accuracy, surpassing alternatives.

- *Claim 3*: FairViT achieves competitive computational efficiency while outperforming alternatives without sacrificing fairness.

To extend the FairViT model, an adversarial debiasing component (Zhang et al., 2018) is proposed for incorporation into the total loss function, reducing the emphasis on sensitive features. Additionally, an improved transparency assessment is introduced.

## 3 FairViT Model

The FairViT model is a novel ViT framework designed to balance fairness and accuracy in computer vision tasks. It belongs to a class of in-processing fairness interventions. It works through embedding fairness constraints directly into the model's training process via adaptive masking 3.1. This improvement in fairness is achieved without requiring training from scratch, external changes to the data, or post-hoc adjustments to predictions. The second innovation is the novel distance loss 3.2 which aims to improve accuracy.

### 3.1 Adaptive Masking

Adaptive Masking refines the attention mechanism by adjusting the model structure to improve accuracy while preserving fairness. This approach assigns group-specific masks and associated weights to the Transformer's attention layers. Rather than manually defining static masks and weights, the model iteratively updates them during training. Consequently, this dynamic process significantly increases accuracy and retains the model's fairness when compared to static masks.

The dataset is first divided according to the two sensitive attributes into $G$ distinct groups. Each sensitive group is then partitioned into $\lfloor G/2 \rfloor$ segments, ensuring that all segments within a group contain the same number of samples, although different groups may have segments of varying sizes. For each segment $i$ (originating from a particular sensitive group), a corresponding mask $M_{l,h,i}$ and weight $\varsigma_i$ are introduced as trainable parameters, where $l$ and $h$ denote the layer and head of the Transformer, respectively. The adaptive masking mechanism modifies the attention computation by incorporating a weighted sum of these masks:

$$\tilde{M}_{l,h} = \sum_{i=1}^{G} \varsigma_i M_{l,h,i},$$

where $\tilde{M}_{l,h}$ is the combined mask applied during attention. The modified attention mechanism for a single head is then computed as:

$$\text{HA}(x, M_{l,h}) = \tilde{M}_{l,h} \odot \text{Attn}(x).$$

The Multi-Head Attention approach of Dosovitskiy et al. (2020) is adopted using concatenation. The mask $M_{l,h}$ and weight $\varsigma$ are updated according to:

$$\frac{\partial L}{\partial M_{l,h,i}} = \begin{cases} \frac{\partial L}{\partial \text{HA}} \cdot \text{Attn}_{l,h}(x) \cdot \varsigma_i, & \text{if } i = g, \\ 0, & \text{otherwise,} \end{cases}$$

and

$$\frac{\partial L}{\partial \varsigma_i} = \begin{cases} \sum_p \sum_d \left( \frac{\partial L}{\partial \text{HA}} \cdot \text{Attn}_{l,h}(x) \cdot \sum_d M_{l,h,i} \right), & \text{if } i = g, \\ 0, & \text{otherwise,} \end{cases}$$

where $g \in \{1, \ldots, G\}$. The update process is accomplished via gradient descent. Backpropagation occurs only through the mask of the group to which the individual is assigned, while the full sum of the masks is still used.

### 3.2 Distance Loss Function

The distance loss function is introduced as a regularizer to enhance classification performance. Instead of relying solely on cross-entropy, FairViT's loss function maximizes the target label's score while minimizing the scores of non-target labels, increasing the likelihood of correct classification. To achieve this, a logistic regression classifier is trained during validation to define a hyperplane that separates correctly and incorrectly classified samples, represented as:

$$\hat{y} + \omega \hat{y}_k + \beta = 0,$$

where $\hat{y}$ is the predicted score for the target label, $\hat{y}_k$ is the cumulative score of the top-$k$ non-target labels, and $\omega$ and $\beta$ are trainable parameters determined during validation. The distance from a predicted score pair $(\hat{y}, \hat{y}_k)$ to this hyperplane is calculated as:

$$\Phi(\hat{y}, \hat{y}_k) = \frac{|\hat{y} + \omega \hat{y}_k + \beta|}{\sqrt{1 + \omega^2}}.$$

During training, the distance loss $L_{\text{dist}}$ encourages correctly classified samples to move further from the decision boundary while pushing misclassified samples closer to it. Formally:

$$L_{\text{dist}} = \begin{cases} -\gamma \Phi(\hat{y}, \hat{y}_k), & \text{if } \hat{y} + \omega \hat{y}_k + \beta \geq 0, \\ \Phi(\hat{y}, \hat{y}_k), & \text{otherwise,} \end{cases}$$

Where $\gamma$ is a non-negative parameter. The overall training objective combines the distance loss with the cross-entropy loss:

$$L = L_{\text{ce}} + \alpha L_{\text{dist}},$$

Where $\alpha$ balances the contribution of the distance loss relative to the cross-entropy component. More in-depth information on the equations and the model can be found in Appendix A.

# 4 Methodology

We used the authors' openly accessible implementation of the FairViT model to reproduce its results. Their code is available at `https://github.com/boweitian/Fair-Vision-Transformer`. However, the outdated code required updates to ensure compatibility with current libraries and frameworks. The CelebA dataset (Liu et al., 2015), as used in the original paper, was utilized for training and evaluation. Furthermore, we incorporated an adversarial debiasing training component and extra transparency methods.

## 4.1 Datasets

The CelebA dataset is a large-scale dataset for facial recognition tasks, containing over 200,000 images, each annotated with 40 binary attributes. To accurately reproduce the results, the same attributes were used: gender and hair color as sensitive attributes, and attraction and expression as target attributes. Following the methodology from the original paper, a script was implemented to partition the dataset as follows: 10% was reserved for testing, as in the original paper. From the remaining 90%, images of the first 80 individuals were included, with a 90:10 split for training and validation.

## 4.2 Experimental setup and code

The original paper evaluated FairViT's performance using the following baseline models: the Vanilla model (Tian et al., 2024), the TADeT-MMD model (Sudhakar et al., 2023), the TADeT model (Sudhakar et al., 2023), the FSCL model (Park et al., 2022), and the FSCL+ model (Park et al., 2022). We reproduced the Vanilla model using the authors' repository. The TADeT-MMD, TADeT, FSCL, and FSCL+ models were not included in their repository.

The original paper deployed the FSCL and FSCL+ models using the open-source implementation provided in Park et al. (2022). To replicate their process, the FSCL and FSCL+ models were also implemented from the original source code. However, training required over 55 hours for the contrastive component on the full dataset. Furthermore, restricting training to only 80 identities led to high accuracy variability across epochs and random seeds, thus we did not fully pursue these for reproduction. The TADeT-MMD and TADeT models remain closed-source. According to the authors of the FairViT model, they themselves implemented the TADeT-MMD and TADeT models from images provided in the original publications. Since no official code was released, it is outside the scope of this paper, leading us to exclude them from our analysis as well.

The Vanilla model is used in the original paper as the unmodified model. However, from their code implementation, we discovered that the skeleton of the FairViT model is actually a pre-trained Data-Efficient Image Transformer (DeiT) (Touvron et al., 2021). Due to uncertainty, we introduce an additional baseline: DeiT, which is also an architecture without fairness components, but shares the same architecture as the FairViT model. DeiT uses a similar patch-embedding mechanism and Transformer backbone as ViT; however, it introduces additional tokens for knowledge distillation. Alongside the standard class token, DeiT incorporates a "distillation token" that the Transformer processes as well. This token interacts with supervision from a teacher model (e.g., a CNN teacher) during training, helping to address data-scarcity issues by leveraging the teacher's inductive biases without requiring massive labeled datasets. Moreover, a bug caused the code to not load the pre-trained architecture for the Vanilla model. To follow the methodology of the original paper we load the pre-trained architecture. For consistency throughout the paper, 'the Vanilla model' refers to the baseline used in the original paper, while 'DeiT' refers to the baseline used in this study.

To verify the authors' first two claims through a quantitative study, we trained the Vanilla model, DeiT and the FairViT model on the CelebA dataset and obtained accuracies and fairness metrics. We also assessed this claim qualitatively, following the authors' approach using the GAR method to analyze model behavior. This involved generating heat maps that highlight the areas contributing to the output, providing insights into the decision-making process. As the code for generating these heat maps was not provided by the authors, we implemented our own method, which is available in our GitHub repository. For our third claim, we tracked the computational efficiency of both models across iterations. The evaluation of the models used the same metrics as in the original paper: Balanced Accuracy (BA), Demographic Parity (DP), and Equalized Opportunity (EO). A detailed description of these metrics can be found in Appendix B.

### 4.2.1 Adversarial Debiasing

The model is extended by incorporating an adversarial debiasing component designed to remove information about the sensitive attributes from the learned representation. Concretely, an auxiliary head $f_{\text{adv}}$ is introduced to predict the sensitive attribute $s$ from the model's internal feature vector. We define the adversarial loss $\mathcal{L}_{\text{adv}}(\hat{s}, s)$ as a standard cross-entropy term:

$$\mathcal{L}_{\text{adv}}(\hat{s}, s) = -\sum_{c=1}^{C_s} \mathbf{1}[s = c] \log(p_{\hat{s}}(c))$$

Where $p_{\hat{s}}(c)$ denotes the predicted probability that $s = c$, $C_s$ is the number of possible values for the sensitive attribute, and the indicator function $\mathbf{1}[s = c]$ ensures that the loss only considers the term where $c$ is the true class label $s$.

During backpropagation, the gradient reversal layer (GRL) inverts the sign of gradients $\nabla \mathcal{L}_{\text{adv}}$ flowing from this auxiliary task back into the feature extractor (Ganin & Lempitsky, 2015). This effectively transforms the adversarial head's objective into a maximization objective for the feature extractor. If the adversarial head produces a prediction $\hat{s}$ with a cross-entropy loss $\mathcal{L}_{\text{adv}}(\hat{s}, s)$, the GRL ensures that the feature extractor receives gradients as the following:

$$-\nabla_\theta \mathcal{L}_{\text{adv}}(\hat{s}, s)$$

As a result, the feature extractor is penalized for any signal that makes $s$ predictable, forcing it to put less emphasis on the sensitive attribute cues within its representation.

A scale parameter $\beta$ controls the relative strength of this adversarial objective. During training, $\beta$ is gradually increased to reduce the emphasis on sensitive attributes. Thus, the scheduling of $\beta$ over the current epoch $t$ and the total number of epochs $T$ determines the balance between overall accuracy and fairness. Four common scheduling approaches were explored: Linear, Exponential, Cosine, and Piecewise (Loshchilov & Hutter, 2016; Li et al.; Kim, 2024; Li & Arora, 2019). We decided on these schedulers because they are commonly used in machine learning and capture different trade-offs in training (Kim, 2024; Shen et al., 2024).

The combined training objective in the FairViT model incorporates the main-task cross-entropy $\mathcal{L}_{ce}$, the distance-based loss $\mathcal{L}_{dist}$, and the adversarial loss $\mathcal{L}_{adv}$. Thus the model is trained to minimize:

$$\mathcal{L} = \underbrace{\mathcal{L}_{\text{ce}}(\hat{y}, y)}_{\text{classification}} + \alpha \underbrace{\mathcal{L}_{\text{dist}}(\hat{y}, y)}_{\text{fairness-driven distance}} + \beta(t) \underbrace{\mathcal{L}_{\text{adv}}(\hat{s}, s)}_{\text{debiasing}}$$

here $\alpha$ is a constant that controls the distance-regularization weight, while $\beta(t)$ scales $\mathcal{L}_{\text{adv}}$ according to one of the four schedulers outlined above. For additional details on the model, please refer to Appendix D.

### 4.2.2 Transparency Method

We implement from scratch the GAR (Abnar & Zuidema, 2022) visualization technique used by the original paper to interpret the model's decision-making. We compute GAR, which weights each layer's attention by the gradient of the output logit with respect to the attention matrix. Specifically, given a target output $\hat{y}$, the gradient $\frac{\partial \hat{y}}{\partial A_l(\mathbf{x})}$ is obtained via backpropagation. We define the per-layer gradient-weighted attention as:

$$\mathcal{A}_l = \begin{cases} A_l(\mathbf{x}) \cdot \dfrac{\partial \hat{y}}{\partial A_l(\mathbf{x})}, & \text{if } l = 0, \\ A_l(\mathbf{x}) \cdot \dfrac{\partial \hat{y}}{\partial A_l(\mathbf{x})} \cdot \mathcal{A}_{l-1}, & \text{if } l > 0. \end{cases}$$

In practice, we (1) average attention across heads, (2) average gradients across the same heads, (3) elementwise-multiply the attention by its gradient, and (4) chain the resulting matrices across layers via matrix multiplication. This chaining step provides a single final heat map highlighting the most salient patches for the model's prediction (Abnar & Zuidema, 2022).

We extend the interpretability methods of the original paper by incorporating a local surrogate interpretability approach inspired by LIME (Local Interpretable Model-Agnostic Explanations) (Ribeiro et al., 2016) to explain our ViTs predictions on facial images. First, each input image was segmented into superpixels using the Quickshift algorithm from scikit-image, producing a small number of meaningful patches that group visually similar regions (e.g., hair, forehead, or eyes).

Next, we generate perturbed samples by randomly turning each superpixel "on" or "off". When a superpixel was turned off, it was zeroed out to simulate its absence. We then passed each perturbed image through the ViT, and the model's predicted probability for the target class was recorded alongside the corresponding binary on/off superpixel pattern.

For each pixel in each image, a coefficient was assigned and averaged across all images to obtain a robust importance map of the model. Since each image is cropped to focus on the celebrity's face, this map should indicate whether the model's attributions align with expectations. This approach aims to avoid cherry-picking particularly useful images and to create a more robust visualization by showing the model's more general behavior on the dataset.

### 4.3 Hyperparameters

To determine the appropriate training parameters, we combined those reported in the paper with the logs provided personally by the authors. Due to inconsistencies across the logs, the parameters mentioned in the paper were given precedence, while the logs served as a backup to verify any missing parameters.

Additionally, the ablation studies from the paper were replicated to determine the optimal hyperparameters, including $\alpha$, $\beta$, $G$, and the function scheduler type $\mathcal{S}$ for adjusting $\beta$ in the adversarial debiasing loss. The search was conducted using grid search, with parameter values taken from the paper and listed in Table 1.

Table 1: Hyperparameter values used in experiments.

| Hyperparameter | Values |
|:---:|:---:|
| $\alpha$ | $\{0, 0.001, 0.01, 0.1, 1\}$ |
| $\gamma$ | $\{0.1, 0.3, 0.5, 0.7, 0.9\}$ |
| $G$ | $\{2, 4, 6, 8, 10, 12, 14\}$ |
| $\mathcal{S}$ | $\{\text{linear, exponential, cosine, piecewise}\}$ |

The optimal hyperparameters are discussed in Section 5, and the complete parameter settings used for reproduction are available in our GitHub repository.

### 4.4 Computational requirements

Ablation studies and adversarial debiasing were conducted on a high-performance computing (HPC) cluster with nine NVIDIA A100 MIG instances, each with 20 GB of HBM2 memory. The reproduction of the original paper's experiments and heatmap generation using GAL and the extended transparency method were performed on a 4070 laptop GPU with 8 GB of GDDR6 memory.

Each experiment, whether an ablation study or adversarial debiasing, was run three times with different seeds, with each iteration taking approximately 40 seconds. Reproducing the original paper's results required around 120 hours while training with adversarial debiasing and conducting ablation studies on different schedulers took approximately 6 hours. We estimate that reproducing the original paper generated approximately 15 kg of $CO_2$, while additional trials contributed another 9 kg. This total emission ($\sim$ 24 kg $CO_2$) would allow a Ferrari SF90 to travel the entire Pacific Coast Highway ($\sim$ 270 km) based on its $CO_2$ emissions per kilometer.

# 5   Results

This section presents the results of our experiments aimed at reproducing the original FairViT model's findings and ablation studies, including the proposed extensions. We also evaluate the reproducibility of specific claims made in the original paper and analyze the performance of the FairViT model when trained with an adversarial debiasing component. The target labels ($Y$) and sensitive labels ($S$) used in the original paper were used in our reproduction. Following the original paper's methodology, each reported result is the mean of three runs with different seeds. However, the original paper did not specify which seeds were used; in our study, we chose seeds 19, 20, and 21.

Our reproduced results are compared to those of the original paper, indicated by a lower $\Delta$ sign. Red numbers denote a decrease in performance, while black numbers indicate an increase. Bold values indicate the highest metric for each evaluation.

## 5.1   Results reproducing original paper

This subsection presents our reproduced results for the Vanilla and the FairViT models on the CelebA dataset and revisits the claims listed in Section 2 to evaluate their validity. As previously mentioned, the authors compared the Vanilla model to the FairViT model, despite their different underlying architectures (ViT vs. DeiT). Therefore, we also examine these claims by comparing FairViT with the DeiT model. However, we accept or reject the claims based on the model used by the authors, the Vanilla model.

To verify *Claim 1*, we conducted the experiments described in the README file of the GitHub repository referenced in the original paper. Our goal was to reproduce the results and evaluate whether the FairViT model improves the fairness metrics, thereby we do not incorporate accuracy scores in the evaluation of this claim. The reproduced results are presented in Table 2.

Table 2: Comparison of fairness scores between Vanilla, DeiT, and FairViT. Appendix F presents the standard deviations of the results across different seeds.

| method | **Y** Attraction, **S**: Gender | | | **Y**: Expression, **S**: Gender | | | **Y**: Attraction, **S**: Hair color | | |
|---|---|---|---|---|---|---|---|---|---|
| | BA$_\%$ | EO$_{e-2}$ | DP$_{e-1}$ | BA$_\%$ | EO$_{e-2}$ | DP$_{e-1}$ | BA$_\%$ | EO$_{e-2}$ | DP$_{e-1}$ |
| Vanilla | 71.69$_\Delta$ 0.67 | 26.30$_\Delta$11.87 | 4.47$_\Delta$ 1.23 | **90.52**$_\Delta$ 1.67 | 6.49$_\Delta$ 1.58 | 1.75$_\Delta$ 0.26 | 74.35$_\Delta$ 0.2 | 5.64$_\Delta$ 2.03 | 1.85$_\Delta$ 0.05 |
| DeiT | **74.05**$_\Delta$ - | **12.45**$_\Delta$ - | **3.71**$_\Delta$ - | 89.56$_\Delta$ - | 4.73$_\Delta$ - | **1.46**$_\Delta$ - | **75.33**$_\Delta$ - | 3.95$_\Delta$ - | 1.66$_\Delta$ - |
| FairViT | 73.01$_\Delta$ 6.95 | 16.50$_\Delta$15.35 | 4.18$_\Delta$ 1.34 | 89.46$_\Delta$ 4.66 | **4.45**$_\Delta$ 2.93 | 1.49$_\Delta$0.285 | 75.04$_\Delta$ 6.52 | **3.47**$_\Delta$ 1.37 | **1.60**$_\Delta$ 0.90 |

While our results differ from those reported in the original paper, particularly regarding the performance of the FairViT model, FairViT generally outperforms the Vanilla model in fairness metrics across experiments, supporting Claim 1. However, we hypothesize that these improvements stem from architectural differences rather than inherent advantages of FairViT. When compared to the DeiT model, the gains in both accuracy and fairness largely disappear.

To verify *Claim 2*, we used the same experimental results used for the evaluation for Claim 1, but without the fairness metrics. Our objective was to reproduce the results and assess whether the FairViT model improves accuracy compared to its alternatives. The reproduced results are presented in Table 3. We observe that the FairViT model achieves accuracy comparable to the Vanilla model, contradicting the findings of the original paper.

In addition to the main performance comparison, an ablation study, as in the original paper, was conducted to evaluate the effectiveness of adaptive masking and distance loss. The results are shown in Table 4. Here, $\Theta$ denotes the adaptive masking method without updating masks and weights, while $\Delta\Theta$ signifies the deployment of adaptive masking with updating masks and weights. The reproduced results indicate that adaptive masking and distance loss do not improve accuracy or fairness metrics, further rejecting *Claim 2*.

Table 3: Comparison of accuracy scores between Vanilla, DeiT, and FairViT. Appendix F presents the standard deviations of the results across different seeds.

| method | **Y** Attraction, **S**: Gender ACC% | **Y**: Expression, **S**: Gender ACC% | **Y**: Attraction, **S**: Hair color ACC% |
|---|---|---|---|
| Vanilla | $76.76_\Delta$ 2.75 | $\mathbf{91.13}_\Delta$ 2.71 | $76.76_\Delta$ 0.28 |
| DeiT | $\mathbf{78.21}_\Delta$ - | $90.16_\Delta$ - | $77.94_\Delta$ - |
| FairViT | $77.96_\Delta$ 6.05 | $90.00_\Delta$ 4.27 | $\mathbf{77.99}_\Delta$ 4.53 |

Table 4: Impact of adaptive masking and distance loss on accuracy and fairness metrics in the FairViT model.

| method | **Y** Attraction, **S**: Gender ACC% BA% $EO_{e-2}$ $DP_{e-1}$ | | | | **Y**: Expression, **S**: Gender ACC% BA% $EO_{e-2}$ $DP_{e-1}$ | | | | **Y**: Attraction, **S**: Hair color ACC% BA% $EO_{e-2}$ $DP_{e-1}$ | | | |
|---|---|---|---|---|---|---|---|---|---|---|---|---|
| $\mathcal{L}_{ce}$ | **78.21** | **74.05** | 12.45 | 3.71 | **90.16** | 89.56 | 4.73 | 1.46 | 77.94 | 75.33 | 3.95 | 1.66 |
| $\mathcal{L}_{ce} + \mathcal{L}_{dist}$ | 78.06 | 73.75 | **11.75** | **3.69** | 90.02 | 89.47 | 4.31 | 1.41 | 77.81 | 75.06 | 4.20 | 1.82 |
| $\mathcal{L}_{ce} + \mathcal{L}_{dist} + \Theta$ | 77.95 | 73.46 | 16.21 | 4.05 | 89.97 | **89.49** | **3.10** | **1.29** | **78.14** | **75.40** | **2.88** | 1.63 |
| $\mathcal{L}_{ce} + \mathcal{L}_{dist} + \Delta\Theta$ | 77.96 | 73.01 | 16.50 | 4.18 | 90.00 | 89.46 | 4.45 | 1.49 | 77.99 | 75.04 | 3.47 | **1.60** |

For *Claim 3*, the Vanilla model was excluded since the gains in accuracy and fairness appear to be a byproduct of architectural differences. We compared the runtimes of the DeiT model and the FairViT model while tracking fairness metrics. During training, the only difference is the inclusion and calculation of the distance loss. On average, the DeiT model completes training in 22.03 seconds, whereas the FairViT model requires 25.53 seconds of training plus an additional 2.76 seconds to compute the distance loss. Despite these extra steps, the FairViT model achieves accuracy levels comparable to the DeiT model, with unstable differences in fairness metrics (see Tables 2 and 3). We find that the added training complexity and increased computational cost may not be fully justified. Thus, we also reject *Claim 3*.

However, examining our implementation of the interpretability tool from the original paper ((Tian et al., 2024)), shown in Figure 1, reveals that for the **Y**: Expression **S**: Gender experiment, the FairViT model focuses more on regions associated with the expression, whereas the Vanilla model spreads its attention more widely. Even if this difference is not captured numerically, other fairness metrics might explain the observed shift in attention. Because of the missing implementation of the visualization by the original authors, our images differ; this discrepancy may result from different choices of images or hyperparameters. We conduct a more robust analysis in Section 5.2.2

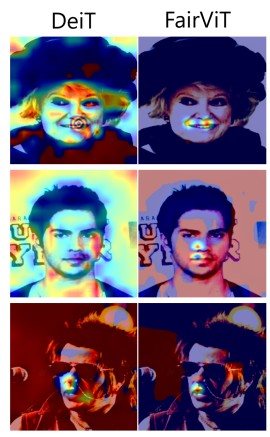

Figure 1: Visualisation of GAR on **Y**: Expression **S**: Gender

### 5.1.1 Ablation Studies

In addition to the main comparison between the performances, we conducted an ablation on the $\alpha$ parameter using the FairViT model, following the methodology from the original paper. The results are shown in Table 5. The original paper reported an optimal $\alpha$ value of 0.01. However, the reproduced results differ from those reported in the original paper and indicate that an $\alpha$ value of 0 may be preferable. This suggests that incorporating the distance loss component $\mathcal{L}_{ce}$ does not lead to a substantial improvement in accuracy scores or fairness metrics, as shown in Table 4.

In addition to the ablation study on the $\alpha$ parameter, we conducted an ablation study on the $\gamma$ parameter using the FairViT model, following the methodology from the original paper. The results are shown in Table

Table 5: Impact of varying $\alpha$ rates on accuracy and fairness metrics in the FairViT model.

| $\alpha$ | **Y** Attraction, **S**: Gender | | | | **Y**: Expression, **S**: Gender | | | | **Y**: Attraction, **S**: Hair color | | | |
|---|---|---|---|---|---|---|---|---|---|---|---|---|
| | ACC$_\%$ | BA$_\%$ | EO$_{e-2}$ | DP$_{e-1}$ | ACC$_\%$ | BA$_\%$ | EO$_{e-2}$ | DP$_{e-1}$ | ACC$_\%$ | BA$_\%$ | EO$_{e-2}$ | DP$_{e-1}$ |
| 0 | **77.99** | **73.48** | 16.14 | 3.94 | **90.19** | **89.71** | **3.39** | **1.40** | 78.01 | **75.73** | 3.79 | 1.73 |
| 0.001 | 77.91 | 73.06 | 14.14 | 4.01 | 86.06 | 84.00 | 10.93 | 2.28 | **78.20** | 75.60 | 3.42 | 1.60 |
| 0.01 | 77.98 | 72.88 | 19.97 | 4.14 | 85.83 | 84.18 | 5.29 | 2.12 | 78.02 | 75.48 | 3.82 | 1.75 |
| 0.1 | 77.48 | 72.43 | **13.07** | **3.62** | 85.67 | 84.13 | 5.88 | 1.92 | 77.31 | 74.88 | 2.53 | **1.53** |
| 1 | 76.68 | 70.24 | 33.01 | 4.80 | 84.02 | 81.58 | 15.23 | 2.46 | 76.60 | 72.40 | **1.40** | 2.90 |

6. The original paper reported an optimal $\gamma$ value of 0.5. However, our reproduced results do not completely coincide with these findings but suggest that the optimal value of $\gamma$ may lie within the range of 0.3 to 0.5.

Table 6: Impact of varying $\gamma$ rates on accuracy and fairness metrics in the FairViT model.

| $\gamma$ | **Y** Attraction, **S**: Gender | | | | **Y**: Expression, **S**: Gender | | | | **Y**: Attraction, **S**: Hair color | | | |
|---|---|---|---|---|---|---|---|---|---|---|---|---|
| | ACC$_\%$ | BA$_\%$ | EO$_{e-2}$ | DP$_{e-1}$ | ACC$_\%$ | BA$_\%$ | EO$_{e-2}$ | DP$_{e-1}$ | ACC$_\%$ | BA$_\%$ | EO$_{e-2}$ | DP$_{e-1}$ |
| 0.1 | 77.92 | 73.46 | 17.33 | 3.98 | 90.05 | 89.52 | 3.59 | 1.40 | **78.14** | 75.56 | 4.39 | 1.77 |
| 0.3 | **78.28** | **73.70** | 19.82 | 4.11 | 90.10 | 89.60 | 3.81 | 1.46 | 78.00 | **75.62** | **1.92** | **1.46** |
| 0.5 | 77.63 | 73.31 | **10.62** | **3.64** | 90.19 | 89.56 | **2.37** | **1.31** | 77.87 | 75.42 | 3.11 | 1.52 |
| 0.7 | 78.01 | 73.67 | 20.09 | 4.11 | **90.31** | **89.74** | 4.63 | 1.58 | 78.10 | 75.21 | 5.26 | 1.81 |
| 0.9 | 77.67 | 73.40 | 14.52 | 3.92 | 90.08 | 89.53 | 4.01 | 1.44 | 77.88 | 75.30 | 3.42 | 1.70 |

Additionally, we conducted an ablation study on $G$ using the FairViT model, following the methodology of the original paper. The original paper reported an optimal $G$ value of 10. Our reproduced results, shown in Figure 2, differ from those of the original paper and suggest that the optimal $G$ value may be closer to 8. Furthermore, our reproduced results indicate that increasing the number of groups beyond this point may negatively influence the performance, potentially due to excessive partitioning of information.

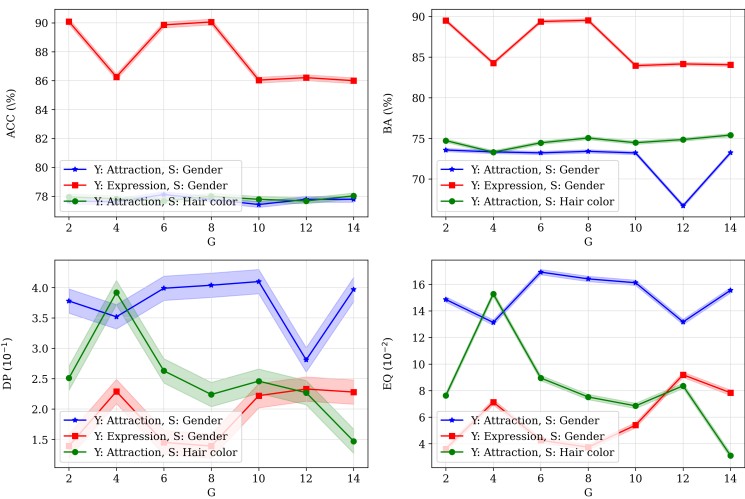

Figure 2: Impact of $G$ on the accuracy and fairness metrics.

### 5.2 Results beyond original paper

### 5.2.1 Adversarial Debiasing

To extend debiasing beyond adaptive masking, we added an adversarial debiasing loss $\mathcal{L}_{\text{adv}}$ to the training loss. This loss is controlled by the $\beta$ parameter, which varies according to one of four schedulers, $\mathcal{S}$. The $\beta$ parameter was modified within the same range $[0, 2]$ for all scheduler types. The linear, exponential, and cosine schedulers followed the growth trends inherent to their respective functions. See Appendix C for the exact functions. For the piecewise scheduler, boundaries were set at 33% and 66% of the iterations: $\beta$ was set to 0 during the first third, 0.67 in the second third, and 1.33 in the final third. Table 7 presents the ablation study results for different $\mathcal{S}$.

Table 7: Impact of a different scheduler $\mathcal{S}$ on accuracy and fairness metrics in the FairViT model.

| $\mathcal{S}$ | **Y** Attraction, **S**: Gender | | | | **Y**: Expression, **S**: Gender | | | | **Y**: Attraction, **S**: Hair color | | | |
|---|---|---|---|---|---|---|---|---|---|---|---|---|
| | $\text{ACC}_\%$ | $\text{BA}_\%$ | $\text{EO}_{e-2}$ | $\text{DP}_{e-1}$ | $\text{ACC}_\%$ | $\text{BA}_\%$ | $\text{EO}_{e-2}$ | $\text{DP}_{e-1}$ | $\text{ACC}_\%$ | $\text{BA}_\%$ | $\text{EO}_{e-2}$ | $\text{DP}_{e-1}$ |
| **linear** | 77.98 | 73.60 | 2.12 | 2.24 | 90.02 | 89.47 | 2.35 | 1.03 | 77.91 | 75.31 | 3.13 | 1.68 |
| **exponential** | **78.04** | 73.58 | 17.45 | 3.97 | **90.25** | **89.67** | 4.12 | 1.51 | 78.04 | 75.28 | 2.76 | **1.19** |
| **cosine** | 76.23 | 72.87 | 3.90 | 2.85 | 89.54 | 89.14 | **1.29** | **0.80** | 76.03 | 73.32 | 2.67 | 1.41 |
| **piecewise** | 78.00 | **73.63** | **1.77** | **2.03** | 89.90 | 89.40 | 1.33 | 1.02 | **78.11** | **75.62** | **1.00** | 1.31 |

Table 7 shows that adversarial debiasing significantly improves fairness metrics while preserving the FairViT model's accuracy. The piecewise scheduler performed best, indicating that a sudden, aggressive increase in $\beta$, after the model has learned from all attributes, enhances performance and effectively reduces bias. We hypothesize that setting $\beta$ at zero early in training allows the model to learn robust discriminative features for the primary task without immediately discarding useful correlations. Later, increasing $\beta$ further reduces the influence of sensitive attributes in the learned representations.

### 5.2.2 Enhanced Transparency Assesment

We further examined the interpretability of both models using LIME. Figure 3 illustrates how LIME highlights important regions in a single sample image, reflecting the model's focus areas. In line with the **Y**: Expression, **S**: Gender experimental setup, the primary region of interest for the model centers around the mouth, an area critical for recognizing facial expressions. However, the model also allocates attention to the nose and chin, regions that potentially encode gender-related information.

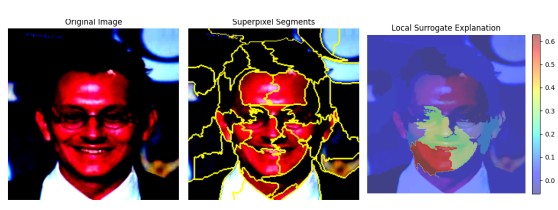

Figure 3: LIME applied to a single image using the FairViT model on **Y**: Expression **S**: Gender

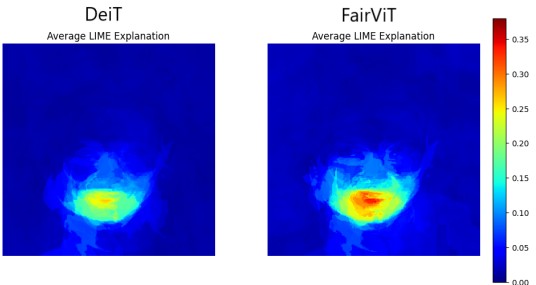

Figure 4: Average LIME Coefficents per pixel across 32 images

By averaging the LIME explanations across 32 images, a clearer pattern emerges, as seen in Figure 4. The FairViT model demonstrates more consistent localization on facial regions relevant to expression, indicating greater stability in isolating target features while minimizing attention to gender cues. This observation

suggests that, despite its additional computational cost, the FairViT model may more effectively target the features necessary for fair and accurate predictions.

### 5.2.3   Optimal Model

Based on our reproduced results, we propose an optimized model called: "Fair Data-Efficient Image Transformer with an Adversarial Debiasing Component" (FairDeiTA). FairDeiTA extends DeiT by incorporating adversarial debiasing with a piecewise scheduler. This model achieves the best performance while maintaining high fairness metrics. Table 8 compares its results with those of the FairViT model, depicting improved results for the fairness metrics, while maintaining accuracy.

Table 8: Comparison of accuracy and fairness scores between the FairViT model and our proposed FairDeiTA model. Appendix F presents the standard deviations of the results across different seeds.

| method | **Y** Attraction, **S**: Gender | | | | **Y**: Expression, **S**: Gender | | | | **Y**: Attraction, **S**: Hair color | | | |
|---|---|---|---|---|---|---|---|---|---|---|---|---|
| | $ACC_\%$ | $BA_\%$ | $EO_{e-2}$ | $DP_{e-1}$ | $ACC_\%$ | $BA_\%$ | $EO_{e-2}$ | $DP_{e-1}$ | $ACC_\%$ | $BA_\%$ | $EO_{e-2}$ | $DP_{e-1}$ |
| FairViT | 77.96 | 73.01 | 16.05 | 4.18 | **90.00** | **89.46** | 4.45 | 1.49 | 77.99 | 75.04 | 3.47 | 1.60 |
| FairDeiTA | **78.04** | **73.76** | **1.51** | **2.31** | 89.93 | 89.34 | **1.18** | **0.85** | **78.09** | **75.66** | **2.68** | **1.53** |

## 6   Discussion

In this study, we present results from various experiments to evaluate the claims made in the original paper (Tian et al., 2024). Claim 1 is validated because the FairViT model's fairness metrics outperform those of the Vanilla model. However, Claim 2 is rejected, the FairViT model achieves lower classification performance than the Vanilla model, and the distance loss function does not improve accuracy. Consequently, Claim 3 is also rejected, as the FairViT model does not offer competitive computational efficiency. Its increased complexity—due to adaptive masking and distance loss does not justify its performance, making it both less efficient and less effective than the Vanilla model. However, when compared against the DeiT model, all three claims are rejected, since DeiT outperforms FairViT in accuracy, fairness metrics, and computational efficiency. One hypothesis is that the hyperplane estimated via logistic regression during validation might be misaligned with the training distribution, especially if the dataset is imbalanced or the model's predictions are highly skewed. Future work could explore more robust strategies for boundary estimation perhaps by integrating end-to-end learned margin constraints or designing a curriculum that dynamically adjusts this margin based on model confidence.

To improve the interpretability of the qualitative heatmap analysis from the original paper, an additional assessment was conducted to better illustrate the FairViT model's effect. Given that large-scale Transformers are increasingly adopted in sensitive domains like healthcare and law enforcement, ensuring fairness is paramount. However, "fairness" itself is context-dependent: some tasks may prioritize demographic parity, while others emphasize equalized odds. Moreover, interpretability is a crucial aspect of ethical deployment. Our additional LIME-based analysis shows that fairness-driven modifications can shift the model's focus to more task-relevant regions (like facial expressions) while reducing the emphasis on sensitive cues (like gender-related features). Such enhanced transparency methods can help practitioners more confidently deploy these models in real-world scenarios. While interpretability is crucial for fair decision-making in computer vision, the current attention-based methods (GAR and LIME) still have known shortcomings, such as potential discrepancies between visual heat maps and the true inner workings of the model. A comprehensive interpretability strategy could involve global model-agnostic explanations (e.g., Shapley values) or causal inference methods, which might better capture how sensitive attributes are leveraged or excluded by the model.

Furthermore, an adversarial debiasing component was added to the total loss function to reduce the emphasis on sensitive targets, enhancing fairness metrics while maintaining performance. To improve the FairViT model's effectiveness, future research could focus on multi-classification and operating with multiple sensitive

attributes. Bias can emerge from the interaction of numerous sensitive factors, for example, race and gender. In addition, real-world applications often need to classify multiple sensitive groups instead of just two fairly. Future work could explore techniques that allow the FairViT model to jointly account for sensitive groups, ensuring fairness across complex applications. Additionally, real-world applications require fairness across multiple groups rather than binary classification. Exploring techniques that enable the FairViT model to jointly account for multiple sensitive attributes could improve fairness in complex settings.

### 6.1 What was easy and what was difficult

Reproducing the results required both a careful examination of the paper's methodology and significant effort in adapting the available code to a functioning workflow. The conceptual ideas and overall architecture were fairly straightforward to understand. The authors provided a clear, layer-by-layer description of the network's structure, allowing us to grasp how the model's components were intended to work together. Additionally, the dataset itself was reasonably well described, and the rationale for focusing on specific data types (e.g., the first 80 individuals or measurement conditions) was clear. These elements offered a good starting point, easing our initial interpretation of the project's scope.

On the other hand, multiple issues made the actual replication substantially more challenging. First, the paper gave no explicit mention of the exact architecture of the Vanilla model, hindering a precise replication of its architecture; we had to infer from the number of blocks and general descriptions that it was likely the **base__16__224** version. If the original authors used a different version it might explain the difference in performance. Second, the codebase did not include an explicit training procedure for the Vanilla model, meaning we had to implement our own approach based on best guesses and indirect references in the documentation. Third, A bug caused the code to load a freshly initialized ViT instead of the pre-trained checkpoint from the ViT paper. Fourth, no dedicated visualization tools were provided, however the clear description in the appendix allowed for an easy reimplementation. Fifth, the environment file did not align with the paper's stated framework requirements (citing PyTorch 2.0 in the text but providing an environment using PyTorch 1.13). This mismatch caused dependency conflicts and required manual resolution and testing to ensure we were replicating the training process under a consistent software stack.

Further complicating matters, the authors mentioned training on 80 individuals, while we only matched their reported performance using the full dataset with the bugged Vanilla model (not pre-trained) and the pre-trained FairViT model based on the DeiT architecture. The results of training the FairViT model on the complete dataset can be found in Appendix E. Their baseline model likewise appeared to use the untrained Vanilla model relying on the full dataset, though the paper suggested it was comparable even when only 80 individuals were used for training. We could not find any mention that the FairViT model is not a pre-trained model however from their code it emerges as such.

### 6.2 Communication with original authors

We reached out to the original authors to ensure an accurate reproduction of the FairVit model. We asked about the dataset split used for training the FairViT and FSCL models to confirm our implementation was aligned with the original approach. Additionally, the author shared their implementation results, allowing us to compare them with theirs and gain deeper insights into their methodology.

Later in the project, it was discovered that these differ from those in the original paper, based on hyperparameters found in the logs the authors later provided.

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

# Appendices

## A   Model details

This section delves deeper on the overall loss function by explaining the three components which make up the loss function. As explained earlier the overall loss function can be broken down into $\mathcal{L}_{\text{base}}(\hat{y}, y)$, $\alpha\mathcal{L}_{\text{distance}}(\hat{y}, y)$ and $\alpha\mathcal{L}_{\text{adv}}(\hat{s}, s)$. Where the first component is depicted as the classification loss, the second component as the distance loss, and the third component as the adversarial fairness term.

**Base classification loss**. This is a standard cross-entropy loss ensuring that the model learns to predict the main label $y$. If $\hat{y}$ is the predicted distribution over classes for sample $i$ and $y_i$ is the ground-truth label, then:

$$\mathcal{L}_{\text{base}}(\hat{y}, y) = -\frac{1}{N} \sum_{i=1}^{N} \log(p_{\hat{y}_i}(y_i))$$

**Distance-based fairness penalty**. This term shapes the model's logit space to reduce certain fairness-driven correlations.

**Adversarial debiasing loss**. This component contains an auxiliary adversarial head predicting $s_i$ from the model's representation. To remove sensitive-attribute information, the gradient reversal layer flips the sign of the adversarial gradient going into the feature encoder, the adversarial head itself tries to minimize cross-entropy.

$$\mathcal{L}_{\text{base}}(\hat{y}, y) = -\frac{1}{N} \sum_{i=1}^{N} \log(p_{\hat{s}_i}(s_i))$$

where $p_{\hat{s}_i}$ is the predicted probability of the correct sensitive label for sample $i$. However, the feature encoder sees the negative gradient of $\mathcal{L}_{\text{base}}$, forcing it to maximize this likelihood's error. In effect, the network "unlearns" how to encode $s$ in its hidden representation.

## B   Evaluation metrics details

- **Balanced Accuracy** (BA) measures the average of true positive and true negative rates across sensitive groups and target labels:

$$\text{BA} = \frac{1}{4} \left( \text{TPR}_{s=0} + \text{TNR}_{s=0} + \text{TPR}_{s=1} + \text{TNR}_{s=1} \right).$$

- **Demographic Parity** (DP) quantifies the disparity in positive prediction rates between sensitive groups, where a lower DP value indicates greater fairness:

$$\text{DP} = |P(\hat{y} = 1 \mid s = 1) - P(\hat{y} = 1 \mid s = 0)|.$$

- **Equalized Opportunity** (EO) measures the difference in true positive rates (TPR) between sensitive groups, with lower values indicating more consistent predictions across groups:

$$\text{EO} = |P(\hat{y} = 1 \mid s = 1, y = 1) - P(\hat{y} = 1 \mid s = 0, y = 1)|.$$

## C   Scheduling approaches details

- **Linear**, linearly interpolates between a small initial value $\beta_{\min}$ and a larger final value $\beta_{\max}$. It offers a straightforward, predictable increase in adversarial debiasing (Kim, 2024).

$$\beta(\text{epoch}) = \beta_{\min} + (\beta_{\max} - \beta_{\min})\frac{t}{T}$$

- **Exponential**, changes $\beta$ exponentially from $\beta_{\min}$ to $\beta_{\max}$. Early iterations only mildly penalize reliance on $s$, then adversarial pressure escalates more rapidly later on (Li & Arora, 2019).

$$\beta(t) = \beta_{\min} \frac{\beta_{\max}}{\beta_{\min}}^{\frac{t}{T}}$$

- **Cosine**, is inspired by cosine annealing schedules commonly used for learning rates, $\beta(t)$ gradually transitions from $\beta_{\min}$ to $\beta_{\max}$ in a smooth, half-cosine pattern, often yielding a more stable mid-phase than linear or exponential schedules (Loshchilov & Hutter, 2016).

$$\beta(t) = \beta_{\min} + (\beta_{\max} - \beta_{\min}) \cdot \frac{1}{2}(1 + \cos(\frac{t}{T}\pi))$$

- **Piecewise**, rapidly changes $\beta$ to predefined constants at certain fractional milestones of training. For instance,

$$\beta(t) = \begin{cases} \beta_1, & t < \alpha_1 T, \\ \beta_2, & \alpha_1 T \le t < \alpha_2 T, \\ \beta_3, & t \ge \alpha_2 T, \end{cases}$$

where $0 < \alpha_1 < \alpha_2 < 1$. Piecewise schedules allow sudden increases (or decreases) in adversarial strength at user-defined intervals, e.g., providing a period of uninterrupted feature learning before focusing on debiasing (Li et al.).

## D   Model training details

---

**Algorithm 1** Pseudo-code of FairDeiTA

---

**Require:** Transformer model parameters $\theta$, training data set $T_t$, validation data set $T_v$, threshold $t$, epoch $E$, and learning rate $lr$.
**Ensure:** The accurate and fair model parameters $\theta^*$.

1: Initialize $M_{l,h,i} = 0$, $L = \infty$, $h = 0$.
2: **while** $h < E$ and $L > t$ **do**                                             ▷ Training Stage
3:     **for** all $x \in T_t$ do **do**
4:         **if** $h = 0$ **then**
5:             $L = L_{\text{ce}}$.
6:             Obtain $\frac{\partial L}{\partial \theta}$ and back-propagate.
7:         **else**
8:             $\hat{s} = f_{\mathbf{adv}}(f(x; \theta))$
9:             $L_{\text{adv}}(\hat{s}, s) = -\sum_{c=1}^{C_s} \mathbf{1}[s = c] \log(p_{\hat{s}}(c))$
10:            $L = L_{\text{ce}} + \alpha \cdot L_{\text{dist}} + \beta(t) \cdot L_{\text{adv}}$.
11:            Obtain $\frac{\partial L}{\partial \theta}$, $\frac{\partial L}{\partial M_{l,h,i}}$, $\frac{\partial L}{\partial \varsigma_i}$ and $\frac{\partial L}{\partial f_{adv}}$ and back-propagate.
12:        **end if**
13:    **end for**                                                                  ▷ Validation Stage
14:    **for** all $x \in T_v$ do **do**
15:        $\hat{y} = f_y(x; \theta)$.
16:        $\hat{y}_k = \sum_{i \in \text{topk}/\{y\}} f_i(x; \theta)$.
17:    **end for**
18:    Update $\omega$ and $\beta(t)$ using logistic regression.
19:    $h = h + 1$.
20: **end while**
21: $\theta^* = \theta$. **return** $\theta^*$.

---

# E   Results full data set

| Method | **Y** Attraction, **S**: Gender | | | |
|--------|------|------|------|------|
| | ACC$_\%$ | BA$_\%$ | EO$_{e-2}$ | DP$_{e-1}$ |
| Vanilla | 74.97$_\Delta$ 0.96 | 67.79$_\Delta$ 4.57 | 37.00$_\Delta$22.57 | 5.35$_\Delta$ 2.10 |
| FairViT | **82.34**$_\Delta$1.67 | **78.90**$_\Delta$ 1.06 | **18.33**$_\Delta$17.18 | **4.05**$_\Delta$ 1.21 |

Table 9: Comparison of model results between Vanilla and FairViT trained on the full dataset for the Attractiveness and Gender experiment.

| Method | **Y** Expression, **S**: Gender | | | |
|--------|------|------|------|------|
| | ACC$_\%$ | BA$_\%$ | EO$_{e-2}$ | DP$_{e-1}$ |
| Vanilla | 87.56$_\Delta$ 0.86 | 86.47$_\Delta$ 2.38 | 11.49$_\Delta$ 6.58 | 1.75$_\Delta$ 0.26 |
| FairViT | **92.19**$_\Delta$ 2.08 | **91.93**$_\Delta$ 2.19 | **2.62**$_\Delta$ 1.1 | **1.44**$_\Delta$ 0.24 |

Table 10: Comparison of model results between Vanilla and FairViT trained on the full dataset for the Expression and Gender experiment.

| Method | **Y** Attraction, **S**: Hair color | | | |
|--------|------|------|------|------|
| | ACC$_\%$ | BA$_\%$ | EO$_{e-2}$ | DP$_{e-1}$ |
| Vanilla | 74.47$_\Delta$ 2.01 | 72.72$_\Delta$ 1.83 | **6.55**$_\Delta$ 2.94 | 1.70$_\Delta$ 0.20 |
| FairViT | **84.00**$_\Delta$ 1.48 | **80.58**$_\Delta$ 0.98 | 4.95$_\Delta$ 2.85 | **1.71**$_\Delta$ 1.01 |

Table 11: Comparison of model results between Vanilla and FairViT trained on the full dataset for the Attractiveness and Hair color experiment.

# F   Standard Deviations

Here we represent the obtained standard deviations found in the results across runs with different seeds. We decided to leave out a table of the standard deviations from the ablation study.

| method | **Y** Attraction, **S**: Gender | | | **Y**: Expression, **S**: Gender | | | **Y**: Attraction, **S**: Hair color | | |
|--------|------|------|------|------|------|------|------|------|------|
| | BA$_\%$ | EO$_{e-2}$ | DP$_{e-1}$ | BA$_\%$ | EO$_{e-2}$ | DP$_{e-1}$ | BA$_\%$ | EO$_{e-2}$ | DP$_{e-1}$ |
| Vanilla | 0.029 | 0.016 | 0.028 | 0.003 | 0.013 | 0.006 | 0.035 | 0.007 | 0.010 |
| DeiT | 0.31 | 1.4 | 0.93 | 0.13 | 1.35 | 0.14 | 0.15 | 1.08 | 0.02 |
| FairViT | 0.88 | 4.07 | 0.38 | 0.08 | 0.65 | 0.06 | 0.12 | 0.67 | 0.12 |

Table 12: Standard deviations from the results depicted in Table 2.

| method | **Y** Attraction, **S**: Gender | **Y**: Expression, **S**: Gender | **Y**: Attraction, **S**: Hair color |
|--------|------|------|------|
| | ACC$_\%$ | ACC$_\%$ ACC$_\%$ | |
| Vanilla | 0.029 | 0.003 | 0.029 |
| DeiT | 0.23 | 0.1 | 0.12 |
| FairViT | 0.23 | 0.13 | 0.36 |

Table 13: Standard deviations from the results depicted in Table 3.

| method | **Y** Attraction, **S**: Gender | | | | **Y**: Expression, **S**: Gender | | | | **Y**: Attraction, **S**: Hair color | | | |
|---|---|---|---|---|---|---|---|---|---|---|---|---|
| | $ACC_\%$ | $BA_\%$ | $EO_{e-2}$ | $DP_{e-1}$ | $ACC_\%$ | $BA_\%$ | $EO_{e-2}$ | $DP_{e-1}$ | $ACC_\%$ | $BA_\%$ | $EO_{e-2}$ | $DP_{e-1}$ |
| FairViT | 0.23 | 0.88 | 4.07 | 0.38 | 0.13 | 0.08 | 0.65 | 0.06 | 0.36 | 0.12 | 0.67 | 0.12 |
| FairDeiTA | 0.07 | 0.44 | 2.0 | 0.27 | 0.07 | 0.07 | 0.72 | 0.4 | 0.06 | 0.45 | 0.76 | 0.14 |

Table 14: Standard deviations from the results depicted in Table 8.

