# OpenReview forum: "Toward Fair and Transparent Vision Transformers: Reproducing FairViT and Introducing FairDeiTA"
_TMLR — Rejected by TMLR_

### Review · Reviewer_ubKc · 2025-03-24

**Summary Of Contributions:**

This work tries to reproduce the FairViT model, which aims to improve fairness in Vision Transformers (ViTs) via adaptive masking and a distance-based loss. The authors find that FairViT does not consistently outperform a properly implemented baseline and attribute prior gains to architectural differences. They improve the original implementation, introduce a more accurate baseline (DeiT), and propose FairDeiTA, which augments FairViT with an adversarial debiasing loss. Through extensive experiments on the CelebA dataset and ablation studies across fairness parameters and schedulers, they show that FairDeiTA achieves improved fairness with comparable accuracy. They also implement interpretability tools (GAR and LIME) to visualize model behavior and provide transparency. All code and analysis are released for reproducibility.

**Audience:**

Yes

**Claims And Evidence:**

Yes

**Requested Changes:**

Please try to address or make more discussions about the weaknesses.

**Strengths And Weaknesses:**

Strengths:
- The authors provide a detailed and well-executed reproduction of the FairViT model, identifying inconsistencies between original claims and actual performance. Clear documentation of experimental setups, code issues, and dataset handling improves transparency and trustworthiness.

- They introduce a model, FairDeiTA, that integrates adversarial debiasing into the ViT training pipeline. They also empirically show that this model improves fairness metrics (Equalized Opportunity, Demographic Parity) with minimal loss in accuracy.

- Implements and extends the GAR visualization and introduces LIME-based explanations, offering a more nuanced understanding of model behavior across sensitive attributes.

Weaknesses:
- The paper identifies confusion regarding the baseline models (Vanilla vs. DeiT), which makes it difficult to perform fair comparisons. The authors should more clearly establish a consistent baseline for all evaluations.

- The authors were unable to reproduce all baseline models from the original paper (FSCL, FSCL+, TADeT-MMD, TADeT), which limits the comprehensiveness of the reproduction study.

- There appears to be confusion regarding whether the original paper used 80 individuals or the full dataset. This ambiguity should be clarified with stronger evidence from the original paper or through more direct author communication.

- While the paper identifies that FairDeiTA improves fairness metrics, there could be more analysis of the inherent trade-offs between fairness and accuracy, including theoretical justification.

---

### Review · Reviewer_oTCN · 2025-03-26

**Summary Of Contributions:**

This paper investigates fairness and transparency in Vision Transformers (ViTs), focusing on the reproducibility of FairViT and proposing a new model, FairDeiTA. The authors reproduce FairViT, a framework designed to mitigate biases in ViTs using adaptive masking and a distance-based loss, and evaluate it on the CelebA dataset. Their findings contradict the original FairViT paper, showing that FairViT does not outperform the baseline Vanilla model in accuracy or fairness metrics. To enhance transparency, they apply interpretability techniques to analyze FairViT's decision-making. Building on these insights, they introduce FairDeiTA to achieve improved fairness metrics while maintaining competitive accuracy.

**Audience:**

Yes

**Broader Impact Concerns:**

Nil.

**Claims And Evidence:**

No

**Requested Changes:**

- Conduct a controlled experiment on the pre-training bug, perform a hyperparameter sensitivity analysis, compare model architectures explicitly, and align their results with the original logs.
- Justify the piecewise scheduler through comparisons, explore $\alpha$ and $\beta$ interactions, and document the tuning process comprehensively.
- Strengthen interpretability claims with quantitative metrics (e.g., average attention scores across images) or a user study.

**Strengths And Weaknesses:**

### Strengths:
- The focus on fairness in ViTs addresses a pressing issue as these models gain traction in sensitive domains (e.g., healthcare, law enforcement), where bias can have significant consequences.
- The attempt to replicate FairViT is a valuable contribution to scientific rigor.
- The usage of GAR and LIME offers a qualitative lens into model behavior, enhancing transparency.

### Weaknesses:
- The authors of the paper attempt to reproduce the results of FairViT and report findings that contradict the original study. However, the authors do not clearly illustrate the root cause of the contradictory results. While they identify possible factors, they stop short of providing a definitive explanation or empirical evidence to support their claims, weakening the credibility of their reproduction effort.
- The paper introduces FairDeiTA, a new model that builds on adversarial debiasing and uses a piecewise scheduler for the hyperparameter $\beta$, which controls the weight of the adversarial loss. The model also involves other hyperparameters, such as
$\alpha$ for the distance loss, to balance different objectives like accuracy and fairness. However, the selection process for these hyperparameters lacks transparency and clear justification, limiting the interpretability of FairDeiTA.
- The paper uses GAR and LIME to analyze model interpretability, claiming that FairViT focuses more on expression-related regions. This claim relies on a single qualitative figure (Figure 1), lacking statistical support or broader validation across multiple samples. This weakens the robustness of their interpretability findings.

---

### Review · Reviewer_qGaS · 2025-06-22

**Summary Of Contributions:**

The paper reproduces results analyzing the accuracy and fairness performance of the Fair ViT model proposed by previous work. The method of the previous work introduces two components: 1) adaptive masking, and 2) difference loss function to enhance fairness. The paper's findings differ from previous work's findings and indicate no significant accuracy or improvement in fairness. An explanation for this divergence in findings provided by the authors is that the improvements observed in previous work come through architectural differences instead of the adaptive masking and difference loss function. However, the paper finds some qualitative improvements due to the Fair ViT method. The areas of attention seem to be more relevant to the tasks considered in the Fair Vit Method. The paper verifies this through some additional qualitative visualization methods.  The paper also implements an additional adversarial debiasing technique to the Fair ViT method and finds improvements in accruacy and fairness methods due to this.

**Audience:**

Yes

**Claims And Evidence:**

Yes

**Requested Changes:**

Suggested changes:
- In the visualization of areas of attention, including a comparison of these areas across sensitive attributes, would be helpful for the different methods compared

- Qualitative analysis of the adversarial debiasing method would be helpful to understand if the quantitative improvements also translate to qualitative improvements

**Strengths And Weaknesses:**

Strengths:
- The methods and claims of previous work are stated clearly
- The implementation and analysis methodology are clearly stated
- The reasons for divergence from previous findings are presented clearly

Weaknesses:
- The tasks on which the qualitative analysis is performed are not very clearly described.
- The qualitative analysis does not convincingly capture fairness analysis. Although it shows that the methods result in more focused attention areas, it is unclear how these areas vary according to the sensitive attributes in the baseline compared to the Fair ViT method.
- The newly proposed adversarial debiasing technique does not seem to be subject to the same qualitative analysis

---

### Decision · Action_Editor_H7B1 · 2025-07-29

**Recommendation:** Reject

**Audience:**

No

**Audience Explanation:**

The main goal of this submission is to reproduce and analyze the results of FairViT, which is a paper published in ECCV' 24, but does not receive much attention or follow-up work. The authors also proposed to use the adversarial debiasing loss to improve FairViT, but this loss has long been used in the literature of algorithmic fairness and is not novel. It is not clear how the findings of this paper would be of interest to TMLR's audience.

**Claims And Evidence:**

No

**Claims Explanation:**

A few important experimental design choices are not clearly stated in the submission:
-   qGaS: The tasks on which the qualitative analysis is performed are not very clearly described
-   oTCN: The authors do not clearly illustrate the root cause of the contradictory results to the origial FairViT paper
-   oTCN: The paper uses GAR and LIME to analyze model interpretability. However, this claim relies on a single qualitative figure (Figure 1), hence lacking statistical support or broader validation across multiple samples
-   ubKc: The authors should more clearly establish a consistent baseline for all evaluations

There is no author rebuttal to the reviews, so the above concerns remain unaddressed.